# Complex Organ Construction from Human Pluripotent Stem Cells for Biological Research and Disease Modeling with New Emerging Techniques

**DOI:** 10.3390/ijms221910184

**Published:** 2021-09-22

**Authors:** Ryusaku Matsumoto, Takuya Yamamoto, Yutaka Takahashi

**Affiliations:** 1Center for iPS Cell Research and Application, Kyoto University, Kyoto 606-8507, Japan; takuya@cira.kyoto-u.ac.jp; 2Department of Diabetes and Endocrinology, Kobe University Graduate School of Medicine, Kobe 650-0017, Japan; 3Institute for the Advanced Study of Human Biology (WPI-ASHBi), Kyoto University, Kyoto 606-8501, Japan; 4Medical-Risk Avoidance Based on iPS Cells Team, RIKEN Center for Advanced Intelligence Project (AIP), Tokyo 103-0027, Japan; 5Department of Diabetes and Endocrinology, Nara Medical University, Nara 634-8522, Japan

**Keywords:** pluripotent stem cell, cellular heterogeneity, cell–cell interaction, organoid, bioengineering, organ on a chip, single-cell transcriptomics, spatial transcriptomics, artificial intelligence

## Abstract

Human pluripotent stem cells (hPSCs) are grouped into two cell types; embryonic stem cells (hESCs) and induced pluripotent stem cells (hiPSCs). hESCs have provided multiple powerful platforms to study human biology, including human development and diseases; however, there were difficulties in the establishment of hESCs from human embryo and concerns over its ethical issues. The discovery of hiPSCs has expanded to various applications in no time because hiPSCs had already overcome these problems. Many hPSC-based studies have been performed using two-dimensional monocellular culture methods at the cellular level. However, in many physiological and pathophysiological conditions, intra- and inter-organ interactions play an essential role, which has hampered the establishment of an appropriate study model. Therefore, the application of recently developed technologies, such as three-dimensional organoids, bioengineering, and organ-on-a-chip technology, has great potential for constructing multicellular tissues, generating the functional organs from hPSCs, and recapitulating complex tissue functions for better biological research and disease modeling. Moreover, emerging techniques, such as single-cell transcriptomics, spatial transcriptomics, and artificial intelligence (AI) allowed for a denser and more precise analysis of such heterogeneous and complex tissues. Here, we review the applications of hPSCs to construct complex organs and discuss further prospects of disease modeling and drug discovery based on these PSC-derived organs.

## 1. Introduction

Traditionally, many studies have used animal models, such as mice, rats, chicken, and zebrafish, to clarify the pathophysiology of human diseases and drug discovery. The pathophysiology of human disorders is shared at least in part with the underlying mechanisms proved in these animal models. However, a number of recent studies have suggested substantial species differences between humans and animals. These differences based on body size, metabolism, immune system, genetic background, and living environment have significant impacts on the disease phenotype and effectiveness of drugs. Human pluripotent stem cells (hPSCs) can proliferate infinitely and differentiate into all three germ layer cells. Due to these properties, hPSCs become a valuable source for human biological studies and disease modeling. hPSCs are grouped into two cell types: embryonic stem cells (hESCs) and induced pluripotent stem cells (iPSCs). hESCs traditionally have been utilized to study human biology, including human development and diseases. In addition to hESCs, the emergence of reprogramming technology to develop hiPSCs has expanded the range of hPSC-based studies. These hPSC technologies now provide tremendous opportunities to study human diseases using human cells [1].

The human body is composed of approximately 37 trillion cells and over 200 cell types [2]. All tissues and organs consist of multiple cell types in the three-dimensional (3D) structure, and the functional interactions between these different cells play an essential role in regulating their functions. Notably, multiple types of cells contribute to the development of many human diseases; therefore, it is difficult to establish a disease model using cell lines. Traditionally, PSC-derived cells have been used for disease modeling, drug discovery, and safety pharmacology studies using 2D monocellular culture; however, most monocellular models are unable to recapitulate human organ functions correctly. Thus, in order to develop better in vitro models, complex multicellular structures that mimic in vivo organs are needed [3].

The hPSCs are theoretically pluripotent, giving rise to every cell type in the human body. Many studies, using various methods, have demonstrated that it is possible to construct PSC-derived complex organ-like structures that contain multiple cell types. While regenerative therapy using hPSC-derived organs is the main purpose of these technologies, human developmental studies and disease modeling should also be crucial applications of the technologies. Accumulating evidence suggests that such complex organ models are superior to single-cell type assays in studying organ function. Several approaches are available to create complex organ models, such as co-culture, organoids, and organ on a chip. 

In this review, we provide an overview of the progress in this field, mainly focusing on organoids of the central nervous system (CNS), research on which has recently progressed most rapidly. We also discuss further prospects of hPSC-derived complex organ technologies and recently developed analytical methods, including single-cell RNA sequencing (scRNA-seq), spatial transcriptome techniques, and artificial intelligence (AI), which are suitable for analyzing complex organs.

## 2. Co-Culture of hPSC-Derived Multiple Cell Types

The traditional two-dimensional co-culture method is a simple way to recapitulate the multiple cell model (Figure 1a). This method provides an easy, scalable, and expandable platform for analyzing the cell–cell interactions.

In CNS, interactions between neurons and glial cells, such as microglia, astrocytes, and oligodendrocytes, play important roles in regulating its function. Dysregulation of glial cells might impair neuronal cell function, resulting in various neuropsychiatric disorders, including Alzheimer’s disease. A two-dimensional co-culture method using hiPSC-derived neurons and astrocytes has been applied to understand the neuronal cell networks in CNS [4,5]. Interestingly, the co-culture of neurons and glial cells promoted further maturation of neurons. Co-culture of iPSC-derived neurons from a patient with Schizophrenia and astrocytes demonstrated synapse deficits, which seem to resemble the pathophysiology.

Another representative example of a two-dimensional co-culture method is the use of immune cells. iPSC-based anti-tumor therapy is one of the most valuable applications of iPSCs. The anti-tumor immune response of iPSC-derived CD8^+^ T cells has been successfully recapitulated [6]. In particular, iPSCs generated from mature T cells retain the same rearranged T cell receptor information [7]. Tumor antigen-specific CD8^+^ T cells can be regenerated infinitely from the T cell-derived iPSCs, indicating their potential clinical applications. These two-dimensional co-culture systems are well established in vitro models that are used to analyze the anti-tumor responses.

Although these two-dimensional co-culture methods provide an easily accessible platform to study cell–cell interactions, it is challenging to recapitulate the pathophysiology of complex diseases.

## 3. hPSC-Derived Organoids

Thus far, PSC-derived embryoid body formation has been a popular in vitro differentiation model. Cells in the embryoid body possess an intrinsic ability to spontaneously differentiate into all three germ-layer cells and organize into functional organs [8]. Technical improvements in the directed differentiation of hPSC-derived three-dimensional culture have yielded organoid technologies (Figure 1b). The term organoid is generally used to represent three-dimensional and miniaturized cellular aggregates containing multiple cell types. To date, various hPSC-derived organoids mimicking human organs have been developed. These systems provide a unique opportunity to study cell–cell and tissue–tissue interactions in vitro.

### 3.1. Organoid Models

Sasai et al. have established several organoids that recapitulate multicellular organs. They studied intrinsic programs that drive autonomous organogenesis, termed self-organization [9]. The first example of a self-organized organoid, which recapitulated cell arrangement and polarity in vitro, is cortical organoid [10]. The generated cortical neurons are actually functional and able to form proper neural connection. They also established the optic cup organoid derived from ESCs [11,12]. Three-dimensional retinal and corneal organoids contain several cell types and show photosensitivity potential, indicating that the organoids have appropriate functions [13,14]. Lancaster et al. also established an hPSC-derived cerebral organoid, which included various brain regions and cell types, such as neuron and glial cells. Cerebral organoids recapitulate the features of human cortical development and produce mature cortical neurons with action potential [15]. Subsequently, several subregions of the human brain have also been successfully modeled using organoid techniques. Midbrain organoids were created from hPSCs. In contrast to dopaminergic neurons generated using two-dimensional methods, midbrain organoids are electrically active and produce dopamine [16,17,18]. Sasai et al. have also established the hypothalamus and pituitary organoids [19,20,21]. During the development process, the interaction between the hypothalamus and pituitary is essential for the pituitary cell differentiation and specification [22]. In the organoid model, both the hypothalamus and pituitary were simultaneously induced in one organoid, which recapitulated their interaction, resulting in the self-organization of pituitary hormone-producing cells. More recently, hypothalamic organoids of the arcuate nucleus have been established [23]. In addition, human inner ear organoids have also been developed as sensory organs. These organoids contain vestibular and cochlear hair cells, which exhibit electrophysiological properties similar to those of native sensory hair cells [24,25].

In addition to CNS organoids, including various regions of the brain and sensory organs, many other organoids have been produced. For example, hiPSC-derived liver organoids, named liver buds, contained multiple cell types, such as hepatocytes and cholangiocytes. Transplantation of the organoid into mice showed maturation into the vascularized and functional liver, which produced albumin and metabolized several drugs [26,27,28]. hPSC-derived organoids that recapitulate multiple parts of the digestive tract, such as the esophagus [29,30], fundus and pylorus regions of the stomach [31,32], intestine [33,34,35], and colon [36,37], have been successfully developed. These organoids exhibited villus-like and crypt-like structures resembling the human digestive tract and also contain multiple cell types, such as enterocytes, goblet, enteroendocrine, and Paneth cells. Lung organoids exhibit rudimentary bronchi-like structures and express alveolar cell markers and surfactant proteins [38,39,40]. Wimmer et al. reported blood vessel organoids derived from human PSCs [41]. These human blood vessel organoids contain endothelial cells and pericytes that self-assemble into capillary networks. hPSC-derived kidney organoids contain rudimentary nephrons surrounded by the renal interstitium and endothelial cells [42,43]. Gene expression profiles of kidney organoids were similar to those in the first-trimester human kidneys [44]. Low et al. reported a protocol for generating vascularized kidney organoids by modulating WNT signaling to control proximal and distal nephron segments. These kidney organoids exhibit structural and functional maturation [45].

### 3.2. Developmental Study Using hPSC-Derived Organoids

Human developmental processes are substantially inaccessible because of limited samples for direct and functional investigation. Moreover, although animal models have been used for such purposes, the molecular and cellular mechanisms underlying human development and developmental disorders remain largely unknown. Therefore, hPSC-derived “self-organizing” organoids are potential substitutes for the direct investigation of developing human tissues and organs.

Recently, in vitro models of early human development have been established using organoid technology. Two groups independently reported the generation of in vitro human blastocyst-like structures, named “Blastoids,” from hPSCs [46] and fibroblasts under reprogramming [47]. These technologies provide a novel platform for studying human embryogenesis. The process of human somite formations, named somitogenesis, has also been studied using hPSCs by recapitulating early mesoderm development. These studies have successfully modeled the human segmentation clock, a key biological concept underlying somitogenesis, which gives rise to the segmental pattern of the vertebrate axial skeleton [48,49].

Among the various organoids reviewed above, cerebral organoids are the most commonly utilized in developmental studies. Surprisingly, cerebral organoids recapitulate the complex process of human cortical development, including characteristic progenitor zone organization with abundant outer radial glial stem cells [10,15] and the epigenomic signatures of the human fetal brain [50]. Single-cell RNA sequencing analysis comparing the cerebral organoids and fetal neocortex revealed the recapitulation of human neocortex development by in vitro organoid development [51]. These pieces of evidence strongly encourage and indicate the usefulness of human cerebral organoids in human developmental studies. Furthermore, comparative studies focusing on the biology of brain development among human and non-human primates have also been reported. These studies revealed the divergence in chromatin accessibility during brain development in human and non-human primate cerebral organoids [52,53].

### 3.3. Central Nervous System Disease Modeling Using hPSC-Derived Organoids

Another important application of hPSC-derived organoids is disease modeling (Table 1). Some neuropsychiatric disorders have reportedly been caused by the dysregulation of neuronal connectivity between diverse neuronal populations, such as glial cells; therefore, a disease model using one neuronal subtype may not be sufficient to reproduce the disease phenotype [54], and multicellular organoids provide better opportunities. Lancaster et al. generated cerebral organoids from iPSCs derived from a patient with microcephaly and demonstrated premature neuronal differentiation, which resembled the disease phenotypes [15]. Moreover, cerebral organoids have been applied for the disease modeling of many central nervous disorders, including Alzheimer’s disease [55,56,57,58,59], frontotemporal dementia [60], Huntington’s disease [61], and Parkinson’s disease [62,63]. We previously established a disease model of congenital hypopituitarism with *OTX2* mutation using patient-derived iPSCs. Hypothalamus–pituitary organoids derived from the patient demonstrated that hypothalamic OTX2 regulated FGF10 expression in the hypothalamus as an underlying mechanism, which was required for pituitary progenitor cell differentiation in a paracrine manner [64].

### 3.4. Other Disease Modeling Using hPSC-Derived Organoids

In addition to the models of congenital and degenerative disorders, these techniques have also been applied to various acquired diseases, such as neoplasms. Ogawa et al. reported a glioblastoma model by CRISPR/Cas9-mediated manipulation of oncogenes and tumor suppressor genes using human cerebral organoids [65]. *HRas*^G12V^ and *TP53*^−/−^ cerebral organoids showed an invasive phenotype resembling glioblastoma.

Kidney organoids derived from a patient with heritable kidney disease caused by the *IFT140* mutation exhibited the shortened tubules and club-shaped primary cilia, in which these phenotypes were restored by CRISPR/Cas9-mediated gene correction [66]. Kidney organoids induced from iPSCs derived from a patient with autosomal dominant polycystic kidney disease (ADPKD) showed increased formation of renal cysts similar to the clinical phenotype of the disease [67]. Leite et al. reported an in vitro model of type 1 diabetes using a co-culture of patient-iPSC-derived β cells and autologous immune cells, demonstrating that T cell activation was restricted to iPSC-derived β cells but not to iPSC-derived cardiocytes or α cells [68]. To model the diabetic vasculopathy, blood vessel organoids were exposed to hyperglycemia and inflammatory cytokines, and thickening of the vascular basement membrane was observed [41].

hiPSC-derived organoids have recently been applied to infectious disease models to recapitulate human-specific responses to viral infection and for drug discovery. Cerebral organoids were used to model Zika virus-associated microcephaly. Zika virus-infected cerebral organoids revealed that the virus infected neural progenitor cells and caused a premature differentiation in neural progenitor cells, leading to progenitor depletion. Neural progenitor cell death caused cortical thinning, resembling Zika virus-associated microcephaly [69,70]. Another study also reported that the Zika virus infection caused cell death in organoids and attenuated organoid growth. Importantly, these phenotypes were not observed in the 2D model but in the 3D model [71]. hiPSC-derived lung organoids (particularly, alveolar epithelial type II (AT2) cells) have been used as a model of SARS-CoV-2 infection [72,73,74]. SARS-CoV-2-infected lung organoids showed interferon-, cytokine-, and chemokine-mediated inflammatory responses [73,74], which helped identify SARS-CoV-2 inhibitors [72].

hPSC-derived organoids have also been used to study drug toxicity in target organs. Morizane et al. demonstrated that kidney organoids could be used to study nephrotoxic drugs, such as gentamicin and cisplatin [75]. hPSC-derived cardiomyocytes were also used to model the cardiotoxicity of doxorubicin, trastuzumab, and tyrosine kinase inhibitors [76,77,78]; for example, doxorubicin caused apoptotic and necrotic cell death, reactive oxygen species production, mitochondrial dysfunction, and intracellular calcium concentration. Interestingly, induced cardiomyocytes derived from a patient who underwent doxorubicin-induced cardiotoxicity were more sensitive to doxorubicin treatment than the control [79]. Liver organoids have been used to study hepatocyte injury caused by aryl alcohol, methotrexate, and acetaminophen [80,81,82]. Notably, non-hepatic cells were also critical in the study of hepatotoxicity; for example, Kupffer cells caused an inflammatory response that increased acetaminophen toxicity to hepatocytes, suggesting the importance of multicellular organoids, in which cell–cell interactions were recapitulated to evaluate drug toxicity.

## 4. Assembling hPSC-Derived Organoids

Since hPSC-derived organoids include multiple cell types, they are suitable materials to analyze the interactions between them. On the contrary, it is challenging to simultaneously induce multiple target cells in the correct position under a single culture condition. Several recent studies have employed the strategies to induce target organoids separately and combine them afterward. Such co-culture methods have been termed “assembloids” [83] and can model the interactions among multiple cell types and organ parts found in complex organs (Figure 1b).

Brain-specific organoids were generated from hPSCs and assembled with other organoids. Birey et al. assembled dorsal and ventral forebrain organoids and demonstrated the migration of interneurons containing glutamatergic or GABAergic neurons into the fetal forebrain in vitro [84]. Xiang et al. reported an in vitro model of neuron projections from the thalamus to the cortex by fusing the thalamus brain organoid and cortical organoid [85].

Workman et al. established an enteric nervous system model of the gastrointestinal tract by combining hPSC-derived intestinal organoids and neural crest cells. Interestingly, neural crest cells migrate into the mesenchyme and differentiate into neurons and glial cells [86]. Combining anterior and posterior gut spheroids differentiated from hPSCs enabled self-organization of hepatobiliary pancreatic organ domains specified at the foregut-midgut boundary organoids [87].

Faustino et al. reported neuromuscular organoids by simultaneously generating spinal cord neurons and skeletal muscle cells [88]. This neuromuscular organoid contains functional neuromuscular junctions supported by terminal Schwann cells.

Ogawa et al. established the glioblastoma model as stated above, which has also been used as a glioblastoma metastasis model by combining tumor-like cells and cerebral organoids [65].

## 5. Tissue Engineering of hPSC-Derived Cells

Another approach to construct complex multicellular organs from hPSCs is tissue engineering (Figure 1c). In particular, many studies have applied tissue engineering to hPSC-derived cardiomyocytes, in which hPSCs were induced into cardiomyocytes and aligned in specific locations combined with a biocompatible extracellular matrix (ECM), such as hydrogel, collagen, and fibronectin, to recapitulate human heart tissue. Such recapitulations allow cardiomyocytes to have more mature properties, providing better opportunities for disease modeling, drug testing, and regenerative medicine [89,90].

Three-dimensional bioprinting of PSC-derived cells has emerged as a technology that creates artificial organs similar to the human body. Rudimentary cardiac constructs have been fabricated using bioprinting and iPSC-derived cardiomyocytes, and these structures function and act as cardiac pumps [91]. Another group has reported bipolar cardiac tissue containing atrial and ventricular cardiomyocytes using a 3D bioprinter. Interestingly, this tissue showed chamber-specific drug responses and gene expression [92].

To form a 3D liver-like structure, Ma et al. reported a triculture model containing iPSC-derived hepatocytes, human umbilical vein endothelial cells (HUVECs), and adipose-derived stem cells in hydrogels. This 3D model showed improved morphology, liver-specific gene expression, and metabolic production compared to the 2D model [93].

In addition to cellular heterogeneity, the ECM also plays an important role in regulating cellular and organ functions. However, most tissue engineering studies utilized a single protein and did not reproduce the complex tissue environment in vivo. Therefore, decellularized porcine tissue has been used for tissue engineering to mimic the in vivo ECM environment. Indeed, hiPSC-derived neurons cultured in a decellularized porcine brain matrix demonstrated upregulation of neuronal markers and improved morphology [94].

These tissue engineering technologies with hiPSC-derived cells will open up new avenues for regenerative medicine and disease modeling.

## 6. Organ-on-a-Chip Technologies

Organoid and assembloid technologies have provided a new platform to analyze tissue-tissue and cell–cell interactions in complex organ-like structures in vitro; however, these culture techniques have several limitations. One of them is that two or more cell types must be maintained under the same culture conditions; therefore, it is not easy to analyze the interactions between multiple organs that require different culture conditions. This is where organ-on-a-chip technologies come in. Organ-on-a-chip technology provides a novel in vitro platform with the potential to reproduce physiological functions of in vivo tissue more accurately than conventional culture models (Figure 1d).

Microfluidic devices integrate multiple cell types in chips with various developmental lineages as complex synthetic human tissues. It is also possible to model complex disorders that involve multiple organs in the human body rather than a single organ. Microfluidic devices are also valuable because they can mimic blood flow, which is essential for the functions of various organs, such as the kidney and liver.

Musah et al. described a method to induce kidney podocytes from hiPSCs and a microfluidic organ on a chip to construct a glomerulus chip that recapitulated the structure and function of glomerular capillaries [95]. In addition, hiPSC-derived hepatocytes were induced into a 3D liver on a chip using organ-on-a-chip technology and biomaterials [96]. The blood–brain barrier (BBB) model was created using a microfluidic device and brain vascular endothelial cells derived from hiPSCs co-cultured with rat astrocytes. This BBB-on-a-chip model recapitulated physiological BBB functions and is expected to be a valuable tool for drug screening [97].

## 7. Analysis Methods

### 7.1. Single-Cell Transcriptomics

Genome-wide quantifications of mRNA expression, transcriptome analyses, are powerful methods for characterizing cellular and molecular properties and functions [98]. Until a few years ago, bulk RNA sequencing has been the main approach to obtain transcriptome data; however, it can only provide an average RNA expression profile of the analyzed samples as a cellular group. Therefore, it was not suitable for the accurate characterization of complex multicellular heterogeneous tissues, such as human organs and hPSC-derived organoids. The advent of single-cell RNA sequencing (scRNA-seq) has revolutionized the transcriptome study. The scRNA-seq has a significant impact on the analysis of multicellular heterogeneous tissues because it enables the precise characterization of cellular composition and gene network regulation. There are two major platforms to obtain single-cell transcriptomics: multi-well sorting method (Smart-seq and BD Rhapsody) and droplet-based method (Drop-seq and 10× genomics Chromium) (Figure 2a). Recently, the scRNA-seq has become a standard experiment in many laboratories due to advances in RNA sequencing technologies and rapid cost reduction. 

The scRNA-seq involves several steps to obtain a single-cell transcriptome profile: (1) isolation of each single-cell from samples, (2) RNA extraction and barcode-labeling, (3) reverse transcription to produce cDNA, (4) cDNA amplification to create libraries for sequencing, and (5) sequencing of the libraries. Each cDNA molecule is barcoded with short sequences to give it unique molecular and single-cell identities.

The scRNA-seq is an effective tool for classifying the cell types included in the tissue of interest, determining the developmental trajectory of each cell lineage, and investigating the interactions between different cell types. To date, many studies regarding hPSC-derived tissues have utilized the scRNA-seq to identify the cell types and analyze the interaction between different cell types, such as ligand–receptor regulatory networks [47,85].

### 7.2. Spatial Transcriptomics

Although scRNA-seq has immense potential for the analyses of the complex tissues, it has several limitations. One of the most important limitations is the loss of spatial information. Each cell receives various signals from the surrounding environments (other cells and interstitium) by various means, including ligand-receptor direct interaction, soluble factors (such as growth factors and cytokines), and mechanical forces. These cell–cell and tissue–tissue interactions play fundamental roles in regulating the biological functions of tissues and cells. Therefore, “spatial information” is important to understand the regulatory mechanisms inside the complex tissue. Several methods have been developed to achieve spatially resolved transcriptomics (Figure 2b).

Laser-capture microdissection is a well-established method for obtaining subpopulations in tissue cells of interest under direct microscopic visualization [99]. The obtained samples can be used for downstream NGS analysis to acquire comprehensive gene expression profiles with spatial information. This method has been applied mainly in studies on tumor heterogeneity. However, sample preparation is cumbersome, which results in low throughput in the sample population and subsequent NGS gene detection. 

In situ barcode sequencing (ISS) methods are also available for the spatial transcriptome analysis and simultaneous quantification of hundreds to thousands of mRNA transcripts [100,101]. There are several modified versions of ISS. Nilsson et al. have developed a new method named hybridization-based in situ sequencing (HybISS) [102]. This method consists of several steps: hybridization of padlock probes for selected RNAs of interest to expressed RNAs within the sample tissue, amplification of circularized DNA (rolling amplification) at the locations of the padlock probes, and detection of the resulting fluorescent DNAs using fluorescence microscopy. This method provides high-resolution mRNA expression data at subcellular levels. However, it is not capable of non-biasedly and comprehensively profiling the gene expression in target samples of interest.

Barcoded solid-phase RNA captures are currently commercially available spatially resolved transcriptomics (spatial transcriptomics (ST) and 10× Visium) [103]. In these methods, tissues are sectioned on glass slides, on which spatially barcoded RNA-binding probes are fixed. Then, the tissue sections are fixed, stained, and imaged. Tissue permeabilization allows RNAs to be released from the tissue and polyadenylated mRNAs to bind to the RNA-binding probes, including poly T. Reverse transcription is then performed to generate cDNA with spatial barcodes. The resulting cDNA library is sequenced using a standard Illumina sequencer. With these methods, comprehensive gene expression profiles of the target tissue are obtained along with spatial information. Nevertheless, the limitations of these methods include limited resolution and size of capture areas. In ST, the size of spatial spots corresponding to identical spatial barcode is a diameter of 100 µm, the spots are arranged with a center-to-center distance of 200 µm, and the size of one capture area is 6.2 × 6.6 mm. [104]. The 10× Genomics company acquired ST in 2018 and developed Visium technology, which provides a spatial resolution of 55 μm in diameter and a center-to-center distance decreased from 200 µm to 100 µm.

There are several other methods of spatial transcriptomics, which are summarized in review articles [105,106,107]. A feasible method should be adapted to analyze multicellular tissue, depending on the cell type, size of the tissue, required resolution, and required depth of sequencing. Recently, a spatial transcriptome with the single-cell resolution has been realized. This spatial transcriptomics provides a promising platform for analyzing internal regulatory networks in complex organs derived from hPSCs.

### 7.3. Artificial Intelligence (AI)

Single-cell or spatial transcriptome analyses for the complex multicellular tissues yield a large quantity of data containing unwanted data such as noise. Such big and noisy data are highly complicated for a human to understand patterns and extract biological meanings; therefore, AI-assisted analysis of omics data has been developed [108]. Recently, AI has been utilized in a wide range of fields, not limited to biology. The classic Chinese board game “Go” is challenging for AI to surpass humans because of its complexity and the massive amount of information. “AlphaGo,” developed by an AI company, Deepmind technologies, is a computer program that combines advanced search trees with deep neural networks. Due to rapid advances in AI technologies in recent years, subsequent versions of AlphaGo have become stronger and stronger, finally beating the world champion Go player in 2017.

DeepMind technologies also developed “AlphaFold,” a deep learning program to predict 3D protein structure from the amino acid sequence [109]. Protein structures are commonly determined using experimental methods, such as X-ray crystallography, cryo-electron microscopy, and nuclear magnetic resonance. However, these techniques are expensive and time consuming. Although researchers have applied various computational methods to predict the protein structure, their accuracy has not reached that in experimental techniques. AlphaFold has been trained on more than 170,000 protein data from a public repository of protein sequences and structures. The 3D structure predictions generated by AlphaFold are far more accurate than previous prediction models. AlphaFold has two major versions, AlphaFold 1 and AlphaFold 2. AlphaFold 1 placed first in the overall rankings of the 13th Critical Assessment of Protein Structure Prediction (CASP) competition in 2018. Furthermore, AlphaFold 2 repeated the placement in the 14th CASP in 2020. AlphaFold is now actually utilized to predict the 3D structure of various proteins, including SARS-CoV-2.

These AI technologies have also been utilized in cell biology in various ways. Huang et al. performed machine learning-assisted modeling using a published mouse adult hypothalamic arcuate nucleus (ARC) dataset as a reference and obtained a human ARC signature to identify a population of predicted ARC cells in hiPSC-derived hypothalamic organoid and neonatal human hypothalamus [23]. Machine learning has also been utilized to predict the drug neural toxicity in hPSC-derived neural organoids. It could correctly classify 9 out of 10 chemicals [110]. Moreover, 3D images of in vitro embryos were evaluated using AI to identify small molecules that promote in vitro embryo formation from hPSCs. With the image recognition method, BMP4 was identified as the best small molecule that promotes embryo morphogenesis [111]. Furthermore, using lung organoids and AI prediction models, the impact of gene mutations on SARS-CoV-2 infectivity was assessed [112].

AI is advancing rapidly, and new analysis algorithms and techniques are being developed continuously. These techniques will be widely used in hPSC-derived multicellular organs to better understand cellular properties accurately.

## 8. Discussion and Future Perspectives

This review summarizes recent studies on constructing complex tissues from hPSCs using organoids, assembloids, tissue engineering, and organ-on-a-chip technologies. These complex structures demonstrate better recapitulation of in vivo organs in terms of morphology, gene expression, and function. In addition to these cell biological methods, there have been advances in scRNA-seq and spatial transcriptomics to analyze cellular heterogeneity. Other front-line technologies, including multi-omics, genome editing, and machine learning, also possess great potential for the analysis of these multicellular organs [113,114,115] (Figure 2c). By combining these techniques, hPSC-based organ construction brings new perspectives into developmental studies, disease modeling, and regenerative medicine. These techniques can also be utilized as precision medicine using patient-derived iPSCs.

Among the multicellular organ constructions using hPSCs, organoid technologies have shown the most rapid advancement and diverse applications. These technologies are now widely used in biological research, disease modeling, drug testing, and drug development/repurposing. However, how faithfully PSC-derived organoids recapitulate the in vivo organ functions has not been elucidated. Although accumulating evidence suggests the usefulness of the organoid technologies [15,50], major limitations include the lack of mature structural organization and the limited tissue size, both of which are direct consequences of a lack of functional vasculature [116]. Several approaches have been used to establish perfused vasculature systems, including co-culture with HUVECs, blood vessel organoids, and microfluidic systems [26,96,117]. These methods might provide better recapitulation of human organs in vitro, which leads to a better chance for biological studies and disease modeling.

In addition to the in vitro construction of multicellular tissues, the in vivo generation of chimeric organs in non-human animals is another approach to construct human cell-derived organs [118]. Kobayashi et al. generated rat pancreas from rat PSCs in a mouse knockout of *Pdx1*, which caused impaired pancreatogenesis [119]. In the same way, it is possible to produce human PSCs-derived organs by introducing hiPSCs into the pig embryo in which critical developmental regulators of target organs are knocked out. The shortage of human organs for transplants is a global problem, which may be alleviated by the generation of chimeric human organs. This chimeric organ technique also provides a better model of the human organs created in vivo. This method may also be helpful to analyze human genetic disorders using patient-derived or genome-edited iPSCs.

The combination of these technologies provides a better recapitulation of human disorders than animal models, which can also reduce the use of laboratory animals. Although several issues in iPSC technologies need to be addressed, iPSC-based construction of complex tissues in vitro will improve our understanding of human diseases and enable safe and cost-effective drug screening.

## Figures and Tables

**Figure 1 ijms-22-10184-f001:**
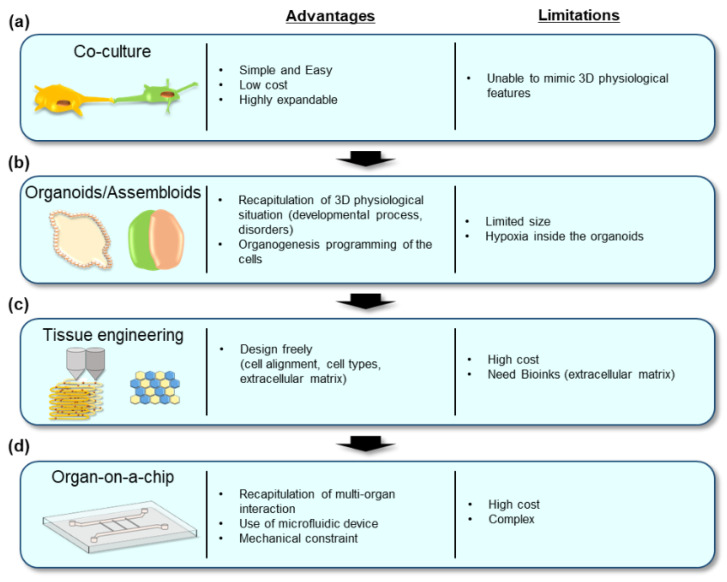
Methods for building multi-cellular models from pluripotent stem cells summarized in this review: (**a**) co-culture method; (**b**) organoids and assembloids; (**c**) tissue engineering; (**d**) organ-on-a-chip technologies. Each method has various advantages and limitations regarding modeling ability, convenience, feasibility, size, biological features, and cost. An appropriate method should be adapted according to the purpose of the study.

**Figure 2 ijms-22-10184-f002:**
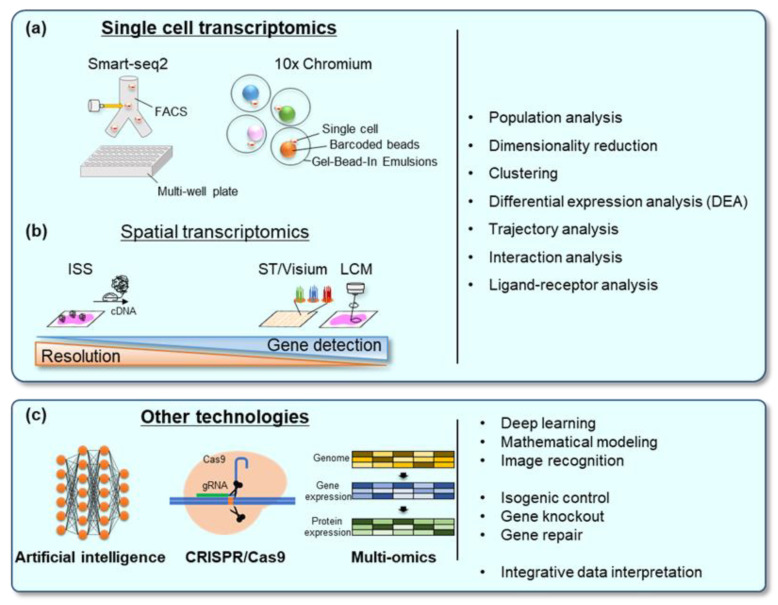
Analysis method for the complex multicellular tissue induced from pluripotent stem cells (PSCs): (**a**) single-cell transcriptomics: several methods have been developed, including Smart-seq2 and 10× Chromium. These methods now play an essential role in analyzing cellular heterogeneity in tissues and organs; (**b**) spatial transcriptomics: these techniques include laser captured microdissection (LCM), in situ sequencing (ISS), and barcoded solid-phase RNA capture (spatial transcriptome (ST) and 10× Visium). Spatial transcriptomics also plays an important role in the study of cellular heterogeneity, especially cell–cell interactions; (**c**) other technologies: artificial intelligence (AI) is now widely utilized in biology to analyze complex data, model construction, and image recognition. CRISPR/Cas9 is also combined with PSC technologies to produce isogenic control cells for patient-derived induced PSCs and edit the gene of interest.

**Table 1 ijms-22-10184-t001:** Summary of various organoids derived from human pluripotent stem cells. Various organoids have been successfully induced from human pluripotent stem cells. This table summarizes representative organoids and includes cell types in each organoid, reports that present the induction methods, and their possible application to human disease modeling.

Organoid Type	Major Included Cell Types	Representative Reports	Possible Applications
Cerebral	Neuron Astrocyte Oligodendrocyte Microglia	Eiraku et al., 2008 [10] Lancaster et al., 2013 [15]	Alzheimer’s disease Microencephaly Multiple sclerosis Epilepsy Frontotemporal lobar degeneration Huntington’s disease Progressive supranuclear palsy Infectious disease (ex. Zika virus)
Midbrain	Dopaminergic neuron Astrocyte Oligodendrocyte Microglia	Tieng et al., 2014 [16] Jo et al., 2016 [17] Monzel et al., 2017 [18]	Parkinson’s disease Multiple system atrophy
Hypothalamus-Pituitary	Pituitary hormone- producing cell Hypothalamic cell	Ozone et al., 2016 [20] Kasai et al., 2020 [21] Matsumoto et al., 2021 [64]	Congenital hypopituitarism Pituitary hormone-producing adenoma Hypophysitis
Inner ear	Vestibular hair cell Cochlear hair cell Sensory neuron	Koehler et al., 2017 [23] Jeong et al., 2018 [24]	Hereditary hearing loss and deafness
Lung	Lung epithelium Alveolar Type I cell Alveolar Type II cell	Gotoh et al., 2014 [38] Dye et al., 2015 [39] Miller et al., 2019 [40]	Cystic fibrosis Hereditary interstitial lung disease Infectious disease (ex. COVID 19)
Liver	Hepatocyte Cholangiocyte Stellate cell Kupffer cell Oval cell	Takebe et al., 2013 [26] Takebe et al., 2017 [27] Guan et al., 2017 [28]	Inherited metabolic liver disease (ex. Wilson’s disease, phenylketonuria, hereditary hypertyrosinemia) Infectious disease (ex. Hepatitis B virus, Hepatitis C virus)
Kidney	Podocyte Endothelial cell Parietal cell Mesangial cell	Takasato et al., 2015 [42] van den Berg et al., 2018 [43]	Autosomal dominant polycystic kidney disease Alport syndrome
Intestine	Enterocyte Goblet cell Paneth cell Enteroendocrine cell Tuft cell	Spence et al., 2011 [33] Watson et al., 2014 [34] Takahashi et al., 2017 [35]	Cystic fibrosis Hirschsprung’s disease Familial adenomatous polyposis Hereditary colorectal cancer

## Data Availability

Data are contained within the article.

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
