# Peer review of "Complex Organ Construction from Human Pluripotent Stem Cells for Biological Research and Disease Modeling with New Emerging Techniques"

_ijms, 2021, doi:10.3390/ijms221910184_

Round 1
Reviewer 1 Report
Figure 1 b) space missing between of and 3D. would also be useful to add microtissue along with organoid/assembloid
This is a well written and clear review.
My only issue is the title matches well with the introduction and discussion, talking about hiPSC and novel techniques.
However, many of your examples jump from hiPSC which was written in full to hPSC (not described) ESC (not described) and even primary cells (HUVEC). It is true that the field is not there yet for fully hiPSC dervied models, so I think that the title is misleading that this review was going to cover specificly hiPSC derived models. Please make a little more clear, and consisitent throughout the review.
Author Response
We appreciate the reviewer‘s valuable comments which significantly improved our manuscript.
Our responses to reviewer’s comments are following;
Following line numbers in the manuscript are presented with “Word Track Changes” on.
Reviewer 1
Comment 1
Figure 1 b) space missing between of and 3D. would also be useful to add microtissue along with organoid/assembloid,
Response 1
We have inserted a space between of and 3D in Figure 1b.
Comment 2
This is a well written and clear review.
My only issue is the title matches well with the introduction and discussion, talking about hiPSC and novel techniques.
However, many of your examples jump from hiPSC which was written in full to hPSC (not described) ESC (not described) and even primary cells (HUVEC). It is true that the field is not there yet for fully hiPSC dervied models, so I think that the title is misleading that this review was going to cover specificly hiPSC derived models. Please make a little more clear, and consistent throughout the review.
Response 2
Thank you for the important comment. As the reviewer pointed, many of our examples presented in the manuscript used not only iPSCs. Therefore, we have revised the manuscript to make it clear as follows
Title: Complex organ construction from human pluripotent stem cells for biological research and disease modeling with new emerging techniques
Line12-16: Human embryonic stem cells (hESCs) have provided multiple powerful platforms to study human biology, including human development and diseases; however, there were difficulties in the establishment from human embryo and ethical issues. The discovery of human induced pluripotent stem cells (hiPSCs) has expanded to various applications in no time because hiPSCs had already overcome these problems.
Line 39-62: Human pluripotent stem cells (hPSCs) can proliferate infinitely and differentiate into all three germ layer cells. Because of these properties, hPSCs become a valuable source for human biological studies and disease modeling. hPSCs are grouped into two cell types: embryonic stem cells (hESCs) and induced pluripotent stem cells (iPSCs). hESCs traditionally have been utilized to study human biology, including human development and diseases. Besides hESCs, the emergence of reprogramming technology to develop hiPSCs has expanded the range of PSC-based studies. These hPSC technologies now provide tremendous opportunities to study human diseases using human cells [1].
Line 88: 2. Co-culture of hPSC-derived multiple cell types
Line 119: 3. hPSC-derived organoids
Line 122-124: Technical improvements in the directed differentiation of hPSC-derived three-dimensional culture have yielded organoid technologies (Figure 1b).
Line 126-127: To date, various hPSC-derived organoids mimicking human organs have been developed.
Line 175: 3.2. Developmental study using hPSC-derived organoids
Line 179-181: Therefore, hPSC-derived “self-organizing” organoids are potential substitutes for the direct investigation of developing human tissues and organs.
Line 204: 3.3. Central nervous system disease modeling using hPSC-derived organoids
Line 221: 3.4. Other disease modeling using hPSC-derived organoids
Line 283: 4. Assembling hPSC-derived organoids
Line 284-285: Since hPSC-derived organoids include multiple cell types, they are suitable materials to analyze the interactions between them.
Line 309: 5. Tissue engineering of hPSC-derived cells
Line 334-335: Indeed, hiPSC-derived neurons cultured in a decellularized porcine brain matrix demonstrated upregulation of neuronal markers and improved morphology [90].
Line 492-494: By combining these techniques, hPSC-based organ construction brings new perspectives into developmental studies, disease modeling, and regenerative medicine.
Line 496-497: Among the multicellular organ constructions using hPSCs, organoid technologies have shown the most rapid advancement and diverse applications. These technologies are now widely used in biological research, disease modeling, drug testing, and drug development/repurposing.

Reviewer 2 Report
The manuscript entitled “Complex organ construction from human iPSCs for biological research and disease modeling with new emerging techniques” seems to be covered most of the topics. The reference list covers relevant literature reflecting the recent developments made in this research area.
I have some minor comments/suggestions:
- Indeed, the major hurdle to studying the pathophysiology of CNS-related disorders is the availability of the representative tissue of the target organ. Recent advances in iPSCs have gained much attention and provide a valuable tool for researchers to study different pathophysiological conditions on a larger scale. My primary concern regarding the usage of iPSCs is how faithfully they recapitulate the target organ. For ex. Pioneering work developed by Sato and Clevers et al. in the field of intestinal organoids faithfully recapitulates the gastrointestinal system, like this, where does iPSC’s stand when it comes to mimicking human physiology and to what extent it is correlated and established.
- It would be very informative to the readers if authors could add a table for the basic composition of media required for organoid growth (Liver, Cerebral organoids, midbrain organoids, hypothalamic and pituitary organoids, inner ear, liver, gut, blood vessels, kidney etc.). Role of each type of organoid in studying human pathophysiology (For example, intestinal organoids: Cystic fibrosis, airway organoids: COVID 19, Cystic fibrosis).
Author Response
We appreciate the reviewer‘s valuable comments which significantly improved our manuscript.
Our responses to reviewer’s comments are following;
Following line numbers in the manuscript are presented with “Word Track Changes” on.
Reviewer 2
The manuscript entitled “Complex organ construction from human iPSCs for biological research and disease modeling with new emerging techniques” seems to be covered most of the topics. The reference list covers relevant literature reflecting the recent developments made in this research area.
I have some minor comments/suggestions:
Comment 1
Indeed, the major hurdle to studying the pathophysiology of CNS-related disorders is the availability of the representative tissue of the target organ. Recent advances in iPSCs have gained much attention and provide a valuable tool for researchers to study different pathophysiological conditions on a larger scale. My primary concern regarding the usage of iPSCs is how faithfully they recapitulate the target organ. For ex. Pioneering work developed by Sato and Clevers et al. in the field of intestinal organoids faithfully recapitulates the gastrointestinal system, like this, where does iPSC’s stand when it comes to mimicking human physiology and to what extent it is correlated and established.
Response 1
We appreciate the reviewer’s important comment. As the reviewer suggested, how faithfully PSC-derived organoids recapitulate the in vivo biological functions of the target organ has not been elucidated. Accumulating evidence suggest the usefulness of the organoid technologies for biological studies, especially developmental studies because some kind of organoids well recapitulate the cellular compositions and gene expression patterns of human embryonic tissue (Camp et al., PNAS 2015). However, in my opinion, it is difficult for in vitro organoid to fully recapitulate in vivo organ’s function because of the lack of fully maturated structural organization. We have added following sentence in Discussion.
Line499-503: However, how faithfully PSC-derived organoid recapitulate the in vivo organ functions has not been elucidated. Although accumulating evidence suggest the usefulness of the organoid technologies [14,46], major limitations include the lack of mature structural organization and the limited tissue size, both of which are direct consequences of a lack of functional vasculature [112].
Comment 2
It would be very informative to the readers if authors could add a table for the basic composition of media required for organoid growth (Liver, Cerebral organoids, midbrain organoids, hypothalamic and pituitary organoids, inner ear, liver, gut, blood vessels, kidney etc.). Role of each type of organoid in studying human pathophysiology (For example, intestinal organoids: Cystic fibrosis, airway organoids: COVID 19, Cystic fibrosis).
Response 2
Thank you for the important suggestion. We agree that a table which present the culture conditions for each organoid will be informative; however, many of induction culture methods consist of several steps which mimic embryonic development and include many signaling molecules. So, it is somewhat difficult to summarize the culture conditions in one table. Instead, we have made a table which summarize representative papers and possible applications for each organoid.
Line 206: Another important application of hPSC-derived organoids is disease modeling (Table).
